# How Plant Polyhydroxy Flavonoids Can Hinder the Metabolism of Cytochrome 3A4

**DOI:** 10.3390/biomedicines13030655

**Published:** 2025-03-07

**Authors:** Carina S. P. Vieira, Marisa Freitas, Andreia Palmeira, Eduarda Fernandes, Alberto N. Araújo

**Affiliations:** 1Associated Laboratory for Green Chemistry (LAQV), Network of Chemistry and Technology (REQUIMTE), Laboratory of Applied Chemistry, Department of Chemical Sciences, Faculty of Pharmacy, University of Porto, 4050-313 Porto, Portugal; carinaspvieira@gmail.com (C.S.P.V.); marysa.freitas@gmail.com (M.F.); egracas@ff.up.pt (E.F.); 2Laboratory of Organic and Pharmaceutical Chemistry (LQOF), Department of Chemical Sciences, Faculty of Pharmacy, University of Porto, Rua de Jorge Viterbo Ferreira, 228, 4050-313 Porto, Portugal; apalmeira@ff.up.pt; 3Interdisciplinary Centre of Marine and Environmental Research (CIIMAR), 4450-208 Matosinhos, Portugal

**Keywords:** polyhydroxy-flavonoids, cytochrome P450 3A4 inhibition, molecular docking, plant–drug interactions

## Abstract

**Background/Objectives**: Recent interest in dietary components and their effects on xenobiotic metabolism has highlighted their role in modulating drug pharmacokinetics. Cytochrome P450 3A4, a key isoform of the cytochrome P450 superfamily, is involved in the metabolism of over 50% of xenobiotics. Flavonoids, present in various foods and supplements, exhibit diverse biological activities influenced by the structural modifications in their scaffold. **Methods**: Fifteen polyhydroxy-flavonoid compounds were firstly tested by a high-throughput fluorimetric method for their ability to inhibit CYP3A4, where scutellarein and gossypetin were assessed for the first time. A molecular docking analysis was performed for the most active inhibitors to gain insight on their interaction with the active site of the enzyme. **Results**: Baicalein, luteolin, and scutellarein were the most potent flavones, presenting an IC_50_ of 15 ± 5, 31 ± 10, and 19 ± 7 μmol/L, respectively. Gossypetin, herbacetin, and quercetin were the most potent flavonols with IC_50_ of 40 ± 8, 32 ± 8, and 23 ± 5 μmol/L, respectively. The molecular docking analysis showed that hydroxyl groups at C6, C7, C8 (ring A), and C3’ (ring B) on the flavone structure affect CYP3A4 enzyme catalysis by binding to its substrate-binding site as strong as known antiviral and antifungal drugs. **Conclusions**: Binding to the enzyme’s active site with a strength comparable to known antifungal and antiviral drugs, baicalein and scutellarein were identified as the most active flavonoids. The vicinal hydroxyls in those molecules were pivotal to positioning and stabilization in the catalytic site pocket.

## 1. Introduction

The cytochrome P450 (CYP450) enzyme family plays a crucial role in the metabolism of endogenous and exogenous compounds. By acting together with other subsequent enzymes of phase II metabolism, they turn molecules more hydrophilic and prone to being excreted [1,2,3].

Human gut and liver primarily express the CYP3A4 isoform, which metabolizes approximately 50% of marketed drugs via the venous portal system [1,4,5,6]. Despite its prevalence, the catalytic performance of CYP3A4 can be unpredictable due to its broad substrate specificity and susceptibility to modulation by innumerable compounds. The behavior of xenobiotics as substrates, inhibitors, or inducers adds complexity to their interactions [1,4,7]. For example, drug–drug or food–drug interactions become especially relevant when the enzyme is inhibited or induced by a drug or nutrient taken concomitantly with another drug [2,3,5]. The inhibition of CYP3A4 can increase the systemic concentration of co-administered drugs, enhancing their pharmacologic effects but also increasing the risk of toxicity, particularly for drugs with a narrow therapeutic index. Conversely, CYP3A4 induction may accelerate drug metabolism, reducing bioavailability and diminishing therapeutic efficacy [8]. The relevance of these events is paramount in situations involving adverse drug reactions and clinical treatment failures [8,9]. Metabolism is also evaluated during drug development because it sheds light on how drugs are processed in the body, their toxicity, how much of them is available in the bloodstream, and their effects on patients [10,11].

Diverse foods and edible natural products contain flavonoids, compounds that share a benzo-γ-pyrone scaffold (C6, C3, C6). The diverse nature of flavonoids has resulted in their sub-classification into flavones, flavonols, flavanols, flavanones, isoflavonoids, chalcones, and anthocyanins [2,12]. Flavonoids are known for their antioxidative, anti-inflammatory, antidiabetic, antiproliferative, anticarcinogenic, and immunomodulatory properties [13]. Soy isoflavones and flavonols found in fruits such as citrus, berries, plums, peaches, grapes, apples, onions, and tomatoes are abundant in the human diet. After hydrolysis, spring onion leaves were found to contain 841 ± 8 mg/kg of quercetin and between 7.84 and 21.30 mg/kg of kaempferol [14,15]. Flavones are found in mandarins, vegetables, apples, mint, celery and grapes, red wine, and propolis [12,13,16,17]. Luteolin, for example, is found between 90 and 137 mg/100 g of the dry weight of oregano and around 848 mg/100 g of the dry weight of tansy leaves [18]. The level of these compounds is affected after food processing. An estimate of the average dietary intake of flavonoids is challenging due to the variety of flavonoids available, their widespread presence in plants, and the diversity in human consumption patterns [12]. Solubility also plays a role in the effect. It is well known that flavonoid aglycones present low water solubility as well as low intestinal absorption, reflecting a modest probability of causing toxic effects [12,16]. However, the ingestion of fat, together with flavonoids, increases the absorption of the compound due to the formation of micelles [16].

Earlier studies have shown that the monooxygenase activity of CYP3A4 can be modulated by flavonoids to varying degrees based on factors like the hydroxyl group count, molecular weight, lipophilicity, and stereostructure [1,19]. Naringenin, for example, inhibits CYP3A4, increasing the systemic concentrations of drugs such as cyclosporine, ranolazine, and methadone [20,21]. To improve drug development and pharmacokinetic prediction, a better understanding of how flavonoids affect CYP enzymes is needed, especially considering the limited research beyond observational studies of CYP450 induction and inhibition [22].

This study aims to systematically evaluate the effects of fifteen hydroxyl-substituted flavonoids, and one unsubstituted flavone (Figure 1)—on CYP3A4 activity. A fluorescence-based enzymatic assay was employed to characterize their inhibitory potential. For the most active compounds, molecular docking simulations were conducted to elucidate their binding interactions with CYP3A4 and to compare their binding affinities with well-known antifungal and antiviral drugs. By integrating experimental and computational approaches, this study provides insights into the molecular mechanisms underlying CYP3A4 modulation, contributing to a better understanding of drug metabolism and potential pharmacokinetic interactions.

## 2. Materials and Methods

### 2.1. Chemicals

The Vivid^®^ CYP3A4 Red Screening Kit (Cat. no. 2856) (enzyme batch 1648816, 1729868 and 2525443) containing human recombinant CYP3A4 Baculosomes^®^, Vivid^®^ BOMR substrate, Vivid^®^ Regeneration system, and Vivid^®^ CYP450 Reaction buffer I was obtained from ThermoFisher Scientific (Carlsbad, CA, USA). Flavone, galangin, resokaempferol, myricetin, and gossypetin were purchased from Indofine Chemical Company, Inc. (Hillsborough, NJ, USA). Kaempferol and scutellarein were obtained from Extrasynthese (Genay, France). Luteolin was obtained from Alfa Aesar (Ward Hill, MA, USA). Fisetin was obtained from Biosynth (Staad, Switzerland). Naringenin, chrysin, apigenin, baicalein, morin, quercetin, herbacetin, and ketoconazole were obtained from Sigma Aldrich, Inc. (St. Louis, MO, USA). Dimethyl sulfoxide (DMSO) was obtained from VWR International (Radnor, PA, USA), and acetonitrile was obtained from Merck KGaA (Darmstadt, Germany).

### 2.2. In Vitro Determination of CYP3A4 Activity

The effect on CYP3A4 catalytic performance was evaluated using the Vivid^®^ CYP3A4 Screening Kit (ThermoFischer Scientific, Waltham, MA, USA), following the indications of the manufacturer. This kit assesses how tested compounds affect the rate of fluorescence increase caused by the breakdown of a resorufin-blocked dye substrate by the human CYP3A4 enzyme. The enzyme promoted the cleavage of substrate at two different bonds, both generating a red highly fluorescent dye, in order to minimize spectral interferences caused by other compounds in the reaction medium. The excitation and emission wavelengths were set to 535 nm and 590 nm with bandwidths of 25 nm and 20 nm, respectively. This setup helped to further circumvent interference from highly fluorescent compounds like NADPH, which are excited at UV wavelengths.

Fifteen flavonoids with hydroxyl substituents and a non-substituted flavone were selected for this study (Figure 1). Stock solutions of the different flavonoids with a concentration of 10 mmol/L were prepared in DMSO.

In a 96-well black flat-bottom assay, the following reagents were added and pre-incubated for 10 min at room temperature: 40 μL of each flavonoid (3.13–50.00 µmol/L) or the positive control (ketoconazole: 0.20–10.00 µmol/L) and 50 μL of a mixture of CYP450 Baculosomes^®^ Plus Reagent and Vivid^®^ Regeneration System. CYP3A4 Baculosomes Plus Reagent^®^ comprised 10 nmol/L recombinant human CYP3A4, human cytochrome *b*5, and cytochrome *c* reductase; 6.66 mmol/L glucose-6-phosphate; and 0.6 U/mL glucose-6-phosphate dehydrogenase in 100 mmol/L potassium phosphate buffer, pH 8.0. The last enzyme and its substrate work as NADPH regenerating system since it catalyzes its formation through NADP^+^, added subsequently. After the 10 min incubation, the catalytic reaction began with the addition of 10 μL of the mixture of 30 µmol/L fluorogen (substrate) and 1000 µmol/L NADP^+^ in potassium phosphate buffer. The fluorescence measurements were performed every 30 s for 30 min in a microplate reader with plate shaking and temperature control, 29 °C (Cytation 5, BIO-TEK, Winooski, VT, USA).

The percentage of inhibition was calculated based on the reaction initial rates measured from the constant slope values of the fluorescence–time charts. An example of a graphic representation is presented in Appendix A. The result for each flavonoid is expressed as a percentage of inhibition (%) ± standard error of the mean (SEM) for at least four independent experiments.

### 2.3. Docking Study of Flavonoids into CYP3A4

The crystal structure of CYP3A4 bound to the known inhibitor ketoconazole was obtained from the Protein Databank (PDB code: 2V0M) [23]. The protein was prepared using AutoDock Tools 1.5.7 by removing additional chains and the crystallographic ligand, adding hydrogen atoms, calculating Gasteiger charges, and merging the nonpolar hydrogens. The 3D structures of the three flavonoids (baicalein, luteolin and herbacetin) were drawn using Hyperchem 7.5 (Hypercube, Gainesville, FL, USA), being minimized by the semi-empirical Polak–Ribiere conjugate gradient method (RMS < 0.1 kcal/Å/mol) [24].

Docking simulations between the target CYP3A4 and flavones and known inhibitors (amiodarone, amprenavir, cimetidine, darunavir, diltiazem, fluconazole, indinavir, ketoconazol, miconazole, nefazodone, and verapamil) were undertaken in AutoDock Vina v1.2 (Scripps Research Institute, San Diego, CA, USA) [25,26]. AutoDock Vina considered the target conformation as a rigid unit, while the ligands were allowed to be flexible and adaptable to the target. AutoDock Vina was employed with an exhaustiveness value of 8, representing the computational intensity, and a grid box spanning X:15.0, Y:19.0, Z:15.0 Å, centered at X: 17.80, Y:8.66, Z: 62.57. This grid box encompassed the substrate binding domain plus the heme group. Autodock Vina calculates its score as the sum of empirically weighted interaction terms—including Gaussian approximations for steric (van der Waals) interactions and hydrophobic and hydrogen bonding contributions—and a torsional penalty that accounts for ligand flexibility. This composite score is designed to approximate the binding free energy (in kcal/mol) of the ligand–receptor complex [26]. A maximum of 9 different conformations were generated for each ligand during the docking run, and the minimum energy pose (higher affinity) was selected for further investigation. The interactions between the active site in the target and the ligand conformations were analyzed using PyMOL (version 1.3).

### 2.4. Statistical Analyses

The statistical analysis comprised both comparisons by the Student’s t-test for mean % inhibition and one-way analysis of variance (ANOVA) for the comparison between concentrations on the IC_50_. Using GraphPad Prism (version 10.2.0), these values were calculated based on the Michaelis–Menten equation through sigmoidal fitting with nonlinear least squares regression. Differences were significant for *p* < 0.05.

## 3. Results

### 3.1. In Vitro Evaluation of CYP3A4 Activity

A panel of different hydroxylated flavonoids and the positive control, ketoconazole, was tested against CYP3A4 to establish a structure–activity relationship. Table 1 shows the % inhibition calculated for each flavonoid included in the study and the IC_50_ values when calculated. The selected compounds were chosen based on previous observations and literature assumptions, featuring up to 6 hydroxyl groups positioned on the A, B and C rings. This selection aimed to investigate the influence of their position in the flavonoid scaffold. The results obtained in this work regarding the in vitro inhibitory activity of CYP3A4 are represented in Figure 2 and showed that baicalein was the most active flavonoid, with an IC_50_ = 15 ± 5 μmol/L. Along with baicalein, the flavones luteolin and scutellarein were also found to inhibit CYP3A4 enzyme, with an IC_50_ of 31 ± 10 μmol/L and 19 ± 7 μmol/L, respectively. In addition, quercetin, herbacetin, and gossypetin were the most active flavonols with an IC_50_ of 23 ± 5 μmol/L, 32 ± 8 μmol/L, and 40 ± 8 μmol/L, respectively. Notably, this study marks the first evaluation of scutellarein and gossypetin. There is only one study on the effect of herbacetin (pollenin A) of *Camellia sinensis* (tea leaves) and flaxseed on CYP3A4 with an IC_50_ < 10 μM [27]. Additionally, the IC_50_ of the positive control, ketoconazole, which was 1.1 ± 0.4 μmol/L, reinforces the precision and dependability of our results. Appendix A represents the impact of the flavonoid substitution of hydrogen by hydroxyl and a double bound between C2 and C3 and the respective statistical significance. Data from the Swiss Institute of Bioinformatics–Swiss ADME (http://www.swissadme.ch; last access 2 July 2024), presented in Appendix A, reveals that the most potent flavones (baicalein, luteolin, and scutellarein) possess three or four hydroxyl groups, which form catechol moieties.

Flavonols present five or six groups that can lead to catechol or hydroquinone portions. However, it was not possible to find robust correlations between the % inhibition and these physical–chemical properties. To verify whether the ionization state of the flavonoids at the experimental pH (phosphate buffer pH 8.0) could explain the results, the theoretical pK_a_ values enabled by the Chemicalize platform (^©^Chemaxon) were used (Appendix A). The most active flavonols (gossypetin and herbacetin) are the ones with the highest pK_a_. The C5 and C7 hydroxyls of A ring in most of the flavonoids are ionized at pH 8.0 and, to a lesser extent, the hydroxyl at C4’ of ring B. Regarding the most potent inhibitors, more precisely flavones, it is not possible to establish a correlation between the pK_a_ and % of inhibition.

In vitro studies using different substrates yield distinct IC_50_ values for the same inhibitor, demonstrating the impact of this aspect [13]. Despite the different substrates used and with chromatography as the preferred technique, comparative values of IC_50_ found in the literature are shown in Table 2.

For baicalein, reported IC_50_ values range from 7.56 and 26.35 μmol/L [5,28,29,30], aligning well with our findings (15 ± 5 μmol/L). Quercetin is a widely studied flavonoid, with different values of IC_50_ being reported in the literature, ranging from 3.03 to 208.65 μmol/L [4,11,30,31,32,33,34,36]. Comparing the results of the present study (23 ± 5 μmol/L) with those found in the literature using the same method but a different substrate (28.0 ± 5.2 μmol/L), we observe high similarity [31]. Regarding luteolin, the previously reported IC_50_ values range from 4.62 to 57.1 μmol/L [4,30,31]. The IC_50_ obtained in this work was 31 ± 10 μmol/L, which stays within the referred interval. Herbacetin showed an IC_50_ < 10 μmol/L as reported by Qian J. et al. [27]. In contrast, our findings indicate an IC_50_ of 32 ± 8 μmol/L, a higher value likely influenced by the differences in the applied methodology. The differences between the IC_50_ values can be attributed to several factors. Different substrates interact with the enzyme with different affinities, leading to distinct IC_50_ values. Structural differences among substrates influence their binding modes, consequently affecting the inhibitory potency. Despite reaction temperature and pH being similar between studies, other conditions were different, such as the human liver microsome concentration ranging between 0.1 and 1.0 mg/mL and incubation times that vary between 0 and 15 min. Minor changes in these parameters can significantly impact the enzymatic inhibition results. Some of the discrepancies observed in the literature are related with several experimental variables, including different substrates and positive controls. While our results align with previously reported ranges, comparing absolute IC_50_ values must always take methodological differences into account.

### 3.2. Interaction with the Active Site of CYP3A4 (Docking Studies)

A docking study using Autodock Vina was conducted to predict the binding modes and affinities of baicalein, herbacetin, and luteolin and to explain the inhibitory effects evidenced before. Several well-documented antifungal and antiviral inhibitors of CYP3A4 were selected as positive controls (amiodarone, amprenavir, cimetidine, darunavir, diltiazem, fluconazole, indinavir, ketoconazole, miconazole, nefazodone, and verapamil), whose structures are represented in Appendix A [8,37]. Ketoconazole was docked to ensure that the enzyme binding site was compatible with the original structure of 2V0M and hence the appropriateness of the chosen docking model [38]. As expected, a high affinity to the enzyme’s catalytic site was obtained, as predicted by the highly negative docking score (−10.8 kcal/mol, Table 3). Moreover, both the docked and crystallographic ketoconazole forms have an RMSD of 0.73 Å (Appendix A). For accurate molecular docking and pose prediction, a threshold value below 2.0 Å is ideal for structure validation.

Corroborating the experimental data, baicalein, herbacetin, and luteolin were able to dock into the active site of CYP3A4 (Figure 3A). By examining interaction potential maps, it is noticed that the aromatic portions of the ligand mainly occur in the hydrophobic regions, whereas the polar groups of the ligands occur mainly on the hydrophilic regions of that map (Appendix A). Several amino acid residues in the CYP3A4 active site are contributing to the binding of the tested compounds. These residues include polar, neutral, and hydrophilic (such as Thr and Ser); polar, acidic, and hydrophilic (Asp); polar, basic, and hydrophilic (Arg); and nonpolar, hydrophobic (such as Ala, Ile, Leu, Met, and Phe) amino acids, which play distinct roles in the binding process (Figure 3 and Appendix A). Polar amino acids are generally more involved in hydrogen interactions, which are essential for maintaining molecular binding stability inside the active site. Conversely, nonpolar amino acids tend to be more prone to establish π–π, π–alkyl, or van der Waals interactions [39].

The binding affinities of the respective top-ranked poses are higher than those of known CYP3A4 inhibitors cimetidine, diltiazem, fluconazole, and verapamil (Table 3). The binding of the three flavonoids to the heme group is predicted to occur through their B or AC ring systems. To gain insight about the binding mode of the top-scored poses, a detailed inspection of the non-covalent interactions was performed (Figure 3B–D; Appendix A).

Baicalein fits the active site with the 5,6,7-trihydroxyl moiety packed against the polar section of the narrow channel near the heme group, which can establish up to four H-bonds with residues Arg-105, Arg-372, Glu-374, and Gly-481. The unsubstituted ring B of baicalein moves deeper into the cavity, and it is stabilized by face-to-face π–π interactions and π–cation interactions with the heme group (Figure 3A,B). Baicalein also forms π–alkyl interaction residues Leu-482 and Ala-370. Furthermore, van der Waals interactions between baicalein and CYP3A4 were also observed with four additional residues (Appendix A).

Herbacetin has a different pattern of substitution on rings A and C (3,5,7,8-tetrahydroxyl); moreover, ring B has a hydroxyl group at position 4’. As it is a wider and longer molecule, it leads to a different top-scored docking pose. It binds through π-stacking interaction between the heme group and rings A and C (Figure 3A,C). A hydrogen interaction is established between the ring C carbonyl group and Thr-309. Ring B establishes parallel-displaced π-stacking interaction with Phe-304 (Figure 3C). Herbacetin also establishes π–alkyl interactions with residues Ile-301, Ala-370, and Leu-482. Additionally, nonconventional C-H hydrogen bonds with Ala-305 are revealed. Moreover, six residues are involved in van der Waals interactions (Appendix A).

The luteolin pattern of substitution originates a docking pose where the ring A and C system once again binds through π–π and π–cation interactions with the heme group, with the hydroxyl at C5 establishing hydrogen interactions with Thr-309 in this case. Furthermore, the hydroxyl groups of ring B establish additional hydrogen interactions with residues Arg-105, Arg-372, and with the heme group (Figure 3D). In addition, π–alkyl interactions are established with residues Ala-305, Ala-370, Leu-482, and the heme group, and van der Waals interactions are established with six additional residues (Appendix A).

Noteworthy, the residues such as Arg-105 [40], Arg-106 [41,42], Ile-301 [43], Phe-304 [44], Ala-305 [39], Ala-370 [45], Arg-372 [41,42], Thr-309 [46], Gly-481 [41], and Leu-482 [47,48] were described as being frequently involved in the binding and stabilization of drugs exhibiting CYP3A4 inhibitory activity. All the tested flavonoids had the C7 hydroxyl of ring A, whose interaction with the enzyme active site was previously hypothesized [30,34]. From the three compounds tested, only baicalein interacted through this hydroxyl with Gly-481.

## 4. Discussion

The structure and substitution of the flavonoid scaffold impacted the activity of the compound toward the enzyme, as noticed before [2,4]. As stated in Appendix A, there is a necessity for the hydrophobicity of the substrate to interact with the active site. This property also impacts the inhibition caused by flavonoids [19]. Second, the absence of hydroxyl groups in the A and B rings stimulates the metabolism of compounds, showing the importance of the referred groups for CYP450 inhibition [19].

The hydroxylation at C7 on ring A has previously been identified as playing a crucial role in the interaction between Fe(III) at the enzyme’s active site and molecular oxygen. [30]. In a consistent way, this is the first substituent group to dissociate [49]. Additional hydroxyl groups at ring A can cause ion–ion interactions, leading to the inhibition of the enzyme. In our study, the % inhibition increase to 28 ± 17% for 50 μmol/L chrysin (*p* = 0.01, when compared with flavone) and up to 72 ± 11% for 50 μmol/L baicalein (*p* = 0.003, compared with flavone), corroborating those previous assumptions. The presence of an extra hydroxyl group at position C6 in compounds like baicalein, a flavonoid found up to 0.9 μg/100 g in Welsh onion (Allium fistulosum) and scutellarein (76 ± 7%), significantly impacted their levels compared to chrysin (28 ± 17%) and apigenin (23 ± 8%), which lack this hydroxyl. In turn, the results obtained for both baicalein and scutellarein evidenced that the hydroxyl at C4’ of ring B is not mandatory for the decrease in CYP3A4 activity since they showed similar inhibition effects, as noticed by others [2,30,34]. As further evidence, for an equally tested concentration of 50 μmol/L, a similar low inhibitory activity was obtained for apigenin and chrysin. Nevertheless, some previous reports pinpoint hydroxylation at C4’ as being important to establish a hydrogen bond with an impact on the inhibition mechanism [30]. In this work, the results for apigenin versus luteolin and scutellarein seem to otherwise indicate that this hydrogen bond does not directly rule strong inhibition but could be related to molecule orientation in the catalytic cage.

Differently, a second hydroxyl group at C3’ at the B ring, as in luteolin (69 ± 7%), which can be found at 0.0015 g–1.03 g/100 g in common oregano (*Origanum vulgare*), enables the formation of a catechol group on that ring that enhanced its ability to inhibit the enzyme when compared with apigenin (*p* = 0.01). The results obtained do not support previous findings that flavanones and flavones with hydroxyl groups at C5 and C7 of ring A and monosubstituted at C4’ of ring B cause a decrease in enzyme activity [50]. A single flavanone-naringenin was tested with the three referred substitutions, and also, no inhibitory effect was noticed. Regarding the most active flavones, baicalein, luteolin, and scutellarein, the same pattern of substitution occurs in the C5 and C7 of ring A. However, baicalein does not present the hydroxyl at C4’ of ring B. In the case of baicalein and scutellarein, ring A forms a gallate adduct, while in luteolin, ring B corresponds to a catechol, both cases with a high affinity for Fe(II) [51,52].

Regarding flavonols, herbacetin and gossypetin had higher activity at the 50 μmol/L concentration, namely with 77 ± 6% and 65 ± 7% inhibition of CYP3A4 activity. The molecular structures show identical substitution patterns on rings A and B, as outlined earlier, except for the C3’ hydroxyl, which is absent in herbacetin. They also differ from the remaining flavonols regarding ring A, which in this case corresponds to a catechol. The presence of a hydroxyl group at C4’ has more impact on the inhibitory effect of CYP3A4 than a methoxy or hydrogen substituent [31]. The results obtained for galangin (4 ± 9% inhibition) and kaempferol (24 ± 8% inhibition) corroborated this observation. Regarding the most active flavonoids-baicalein, scutellarein, herbacetin, luteolin, quercetin, and gossypetin-they differ in the number of hydroxyl groups in ring B. In more detail, baicalein does not have hydroxyl groups in this ring, and scutellarein and herbacetin present one hydroxyl, while luteolin, quercetin, and gossypetin molecules possess two hydroxyls. This can suggest that the pattern of substitution in this ring is not as impactful as in the other rings, i.e., ring A. Thus, the activity and the effect of the flavonoid are related to the spatial distribution of the functional groups and not to the common scaffold shared by these groups of compounds [6].

Furthermore, non-substituted flavonoids at ring C, for example, flavones, can be prone to epoxidations [2,30]. These reactions generate reactive intermediates responsible for enzyme inactivation. The presence of a double bond between C2 and C3 of ring C is related to higher inhibition as well [2,30,34]. Comparing the inhibitory effects of apigenin (23 ± 8%) and naringenin (−11 ± 5%), our results indicated that the presence of a double bond between C2 and C3 in apigenin significantly impacted enzyme catalysis (*p* = 0.01).

The results for non-substituted flavone indicate it could act as an activator of the enzyme. In a previous study [19], this flavonoid increased the formation rates of the two metabolites of midazolam, corresponding to an activation of the enzyme. In another work, an activation of 94.65 ± 7.10% for flavone was noticed [53].

As stated previously, hydroxyl groups play a critical role in hydrogen bonding, which directly affects binding affinity, stability, and inhibition potency. Moreover, Figure 1 highlights the substitution pattern with different hydroxyl groups that can increase the number of hydrogen bond donors and acceptors and lead to stronger interactions with key residues in the CYP3A4 active site. However, the hydroxyl distribution pattern on the flavonoid scaffold can also impact ligand orientation and binding efficiency.

The enzyme active site is composed of an iron-heme group, which is prone to originate reactive oxygen species along the catalytic cycle through hydrogen peroxide and autooxidation shunts [51]. Fe(II) can interact with deprotonation sites as well as other positions with resonance effects [54]. Flavonoids have a catechol core that has an affinity for iron as well as a keto group in the vicinity of a hydroxyl [51,54]. Both structures explain an electron-donating capacity related to the antioxidant activity [51]. Concerning CYP450 catalytic cycle, the iron heme goes through different oxidation states in order to oxidize the substrate. In this process, the rate of substrate oxidation depends on the concentration, substrate levels, oxygen levels, and also pH. Polyphenols with gallol groups show a higher iron oxidation rate than polyphenols with catechol groups. Moreover, compounds with different iron binding possibilities show a lower oxidation rate due to competition between the different binding sites [52]. The presence of a hydroxyl group at either C3 or C5, along with the C4 keto group in the C ring and the C2–C3 double bond, enhances Fe(III) reduction ability [55]. In this work, the most active compounds had a catechol or a pyrogallol moiety. Flavones, more specifically luteolin, have a catechol on the B ring and baicalein and scutellarein a pyrogallol on the A ring. Flavonols, such as quercetin, have a catechol on the B ring and gossypetin and herbacetin on the A ring. These portions might influence the interaction with the enzyme.

Most drugs administered through an oral route are absorbed primarily in the small intestine. Flavonoid aglycones, characterized by high hydrophobicity and low molecular weight, can readily permeate villous epithelial cells in the intestinal lining via passive diffusion. Variations in absorption rates among flavonoids are attributed to structural differences, such as hydroxylation patterns and glycosylation, as well as to pH variations across intestinal segments, which can impact flavonoid solubility, ionization state, and membrane permeability. The absorption of flavonoid aglycones is strongly influenced by the number of hydroxyl groups in their structure and the conjugation position of the B ring. These factors collectively determine the bioavailability of flavonoid compounds in the body. In the gut as well as when reaching the liver, flavonoids are mainly oxidized by CYP450 enzymes [18,56,57]. CYP3A4 is mainly located in organs of the digestive tract linked to nutrient absorption and possesses a large active site that can hold different substrates or a substrate and an inhibitor at the same time, underscoring the need to consider all related metabolic pathways. In vivo, this can influence the binding of endogenous substances and xenobiotics and subsequent effects during metabolism [1,2,58].

The pocket containing the active site of CYP3A4 has a volume of 1386 Å^3^ [59,60]. The ligand-accessible volume, about 520 Å^3^, was characterized by a narrow, more horizontal shape [52,54]. The heme pocket of CYP3A4 is hence very flexible [61] and able to host various types of ligands [62]. Such ligands include macrolide antibiotics (such as clarithromycin and erythromycin), anti-HIV agents (such as ritonavir and delavirdine), antidepressants (such as fluoxetine and fluvoxamine), calcium channel blockers (such as verapamil and diltiazem), several herbal constituents (such as bergamottin and glabridin), amongst others [63]. CYP3A4 inhibitors are, in turn, capable of diverse binding modes [64]. A protein–ligand interaction fingerprint analysis (PLIF) of X-ray diffraction data from CYP3A4 structures bound to diverse inhibitors revealed 31 residues involved in hydrogen bonds or π–π interactions (Appendix A). Ser-119 is the most frequent amino acid (63.5%), but other residues are involved in polar interactions, such as Phe-57, Arg-105, Arg-106, Phe-108, Arg-212, Phe213, Asp-217, Phe-220, Thr-224, Val-240, Ph3-304, Ala-305, Thr-309, Ile-369, Ala-370, Arg-372, Cys-442, Gly-481, and Leu-482 (Appendix A).

Molecular docking has been widely used to investigate the binding mode of CYP3A4 with its modulators and substrates [65,66,67] as well as the binding mechanism of flavonoids to several targets [68,69,70].

Several flavonoids (quercetin, hyperoside, isoquercitrin, quercetin-7-*O*-glucoside, rutin, and quercetin-3-*O*-sophoroside) have recently been described as being able to form stable complexes with CYP3A4 by means of hydrogen bonds and van der Waals forces established with the heme-containing active site, which was determined by spectroscopy and computer simulations [71]. Molecular docking revealed that van der Waals forces and π–π interactions were primarily responsible for the CYP3A4–flavone interaction [55]. According to a QSAR study performed by Li et al. [5], a bulky group or hydroxyl groups at C6 and C7 in ring A enhance the inhibitory capacity of the compound. In this respect, four of the five most active inhibitors present three substituents on ring A, creating a bulky group of substituents, and two of the three most inhibitory flavones present a hydroxyl in the A ring of C6.

Flavonoid interactions are driven primarily by the number and diversity of non-covalent interactions (hydrogen bonding, π–π stacking, hydrophobic interactions), as recent molecular docking studies for some compounds of the class show [58]. The interactions with the flavonoids astilbin, isoastilbin, and neoastilbin were clarified using spectral analysis, molecular docking, and molecular dynamics simulation and ascribed to several hydrogen bonds and van der Waals forces [72]. Being capable of such different interactions, it was noticed that quercetin displaces CDK inhibitors from the CYP3A4 binding domain. Its phenyl group docks near Ser-119 and enables the establishment of polar interactions with this residue and Arg-212 [11,73]. Okanin, a primary flavonoid found in a common herbal tea interacts with the binding site, forming three hydrogen bonds with Arg-105, Phe-215, and Glu-374 [74]. As far as naringin is concerned, its hydroxyl groups are positioned in the CYP3A4 binding pocket to facilitate polar interactions with Arg-105, Ser-119, Arg-212, Ser-312, and Gln-484. Cation–π bondings are further established with Phe-108, Phe-213, and Phe-215 [11]. Pelargonidin, a CYP3A4 inhibitor, interacts with the CYP3A4 aromatic rings Phe-447 and Phe-435 via π–π stacking with the flavonoid’s A ring [75]. Auriculasin, a polyphenol flavonoid, binds to CYP3A4 residues Arg-106, Arg-372, and Glu-374 through hydrogen bonds [76]. Hence, CYP3A4 inhibition by flavonoid derivatives can occur through different binding modes, and the number and position of hydroxyl and/or methoxyl groups influence the inhibitory activity of this class of molecules towards the enzyme [77].

The molecules analyzed by docking (baicalein, herbacetin, and luteolin) possess 0, 1, or 2 hydroxyls on ring B, and while inhibition was noticed, the orientation of ring B or rings A and C differs. With baicalein, the former directly interacts with the heme portion through π–π and π–cation interactions. In herbacetin, this ring interacts with Phe-304 at the α-helix I through π–π interactions and not by the hydroxyl present on C4’. On the other hand, luteolin interacts through ring B hydroxyls with the farthest Arg-105 and Arg-372 through hydrogen interactions. In line with % inhibition and docking results, it becomes clear that every ring has a role in the interaction with the enzyme, and depending on the compound, the interactions are different. Besides this, the rotation of ring B contributes to the accommodation of the molecule in the active site.

## 5. Conclusions

In this study, a panel of 16 flavonoids (6 flavones, 9 flavonols, and 1 flavanone) was investigated regarding their inhibitory activity towards CYP3A4, an enzyme responsible for the metabolism of xenobiotics, including drugs, foods, and pollutants. From the tested compounds, five flavonoids showed an enzyme inhibition above 50%. Binding to the enzyme’s active site with a strength comparable to known antifungal and antiviral drugs, baicalein and scutellarein were identified as the most active flavonoids. Additionally, the substitution pattern has an influence on the obtained % of inhibition, and docking studies confirmed the role of the hydroxyl substituents on the molecule’s accommodation to the active site.

The need for drug efficacy is a concern of the pharmaceutical industry and is important in drug development. Given the current emphasis on the high nutritional benefits and low environmental impact of plant-based foods, the pattern of adverse food-drug interactions is evolving. However, these interactions must still be considered, especially due to the increasing prevalence of polymedication, particularly among the elderly. Including natural products in our diet, rich in compounds that can impact enzymatic function and metabolism, should be a focus of pharmacokinetic and pharmacodynamic studies. Despite low bioavailability, flavonoid metabolism is not limited to the liver, and the intestinal mucosa, kidneys, and other tissues also play significant roles in processing and modifying flavonoids, contributing to their overall metabolic profile in the body. High-throughput screening techniques, like the one used in this work, have proven effective for evaluating their impact on CYP450 metabolism.

## Figures and Tables

**Figure 1 biomedicines-13-00655-f001:**
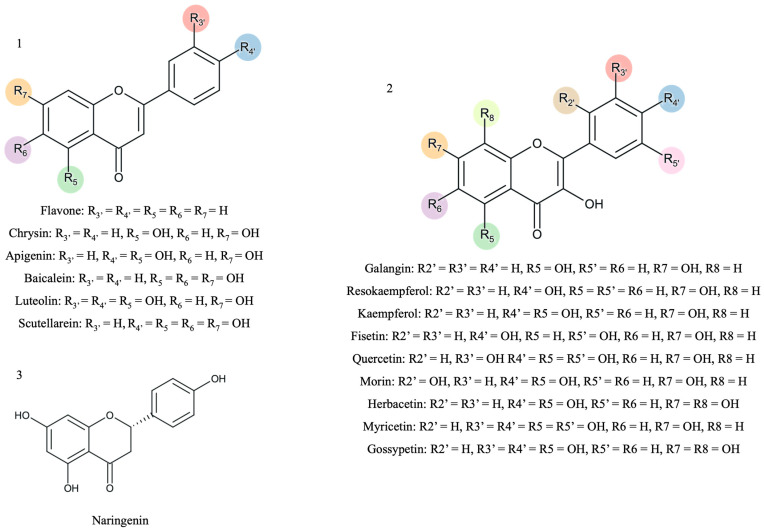
The scaffold representation of the studied flavonoids (1—flavones; 2—flavonols; 3—flavanones). The colors represent the different possibilities of substitution with hydroxyl groups.

**Figure 2 biomedicines-13-00655-f002:**
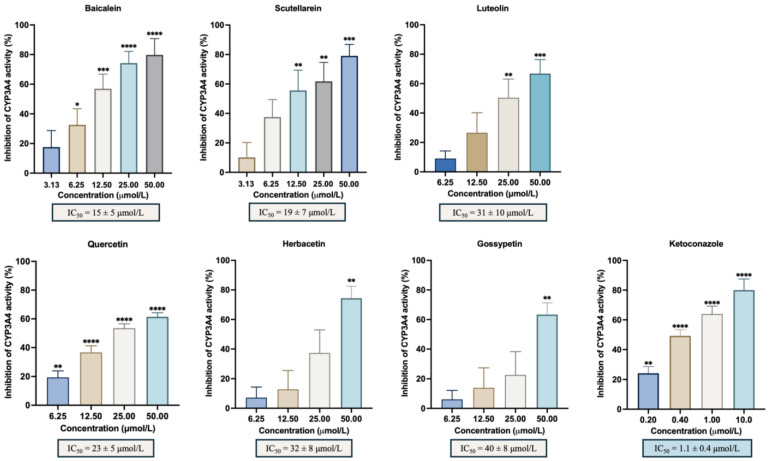
In vitro inhibitory activity of CYP3A4 by the flavonoids with % of inhibition ≥50%: baicalein, luteolin, scutellarein, quercetin, herbacetin, gossypetin, and the positive control, ketoconazole. The results are expressed as the mean of the % inhibition ± SEM and represent at least four experiments. * *p* < 0.05; ** *p* < 0.01; *** *p* < 0.001; **** *p* < 0.0001 when compared with the control (with enzyme and without the compounds under study).

**Figure 3 biomedicines-13-00655-f003:**
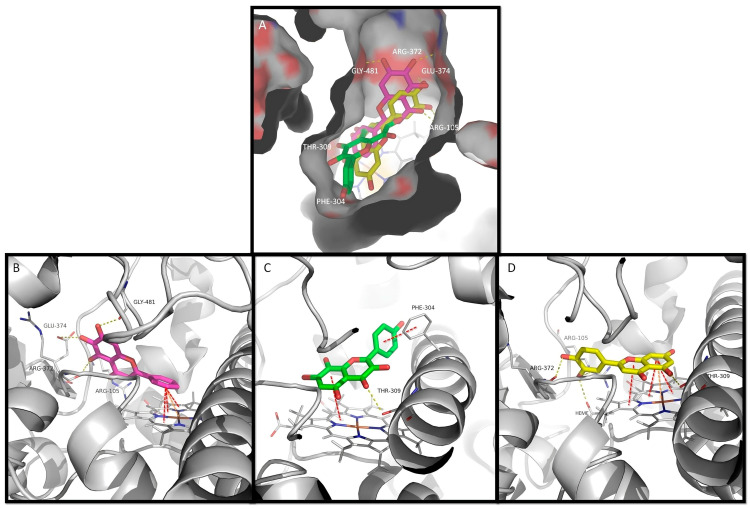
The docking of flavones into CYP3A4. (**A**) The surface representation of CYP3A4 active site and the docked flavones baicalein, herbacetin, and luteolin represented as pink, green, and yellow sticks, respectively. A detailed view of top ranked poses of (**B**) baicalein (pink sticks), (**C**) herbacetin (green sticks), and (**D**) luteolin (yellow sticks) on the active site. Hydrogen interactions, π–π interactions, and cation–π interactions are represented as yellow, red, and orange broken lines, respectively, and the involved residues are represented as thin sticks that are labeled.

**Table 1 biomedicines-13-00655-t001:** Chemical structures of the studied flavonoids, the positive control, and their in vitro inhibitory activity (%) against CYP3A4 at 50 µmol and IC_50_ (mean ± SEM).

Compound	Ring A				Ring B				CYP3A4 Inhibition (%) (50 µmol/L)	IC_50_ (µmol/L)
	R_5_	R_6_	R_7_	R_8_	R_2’_	R_3’_	R_4’_	R_5’_		
Flavone	H	H	H	-	-	H	H	-	−179 ± 73	-
Chrysin	OH	H	OH	-	-	H	H	-	28 ± 17	-
Apigenin	OH	H	OH	-	-	H	OH	-	23 ± 8	-
Baicalein	OH	OH	OH	-	-	H	H	-	72 ± 11	15 ± 5
Luteolin	OH	H	OH	-	-	OH	OH	-	69 ± 7	31 ± 10
Scutellarein	OH	OH	OH	-	-	H	OH	-	76 ± 7	19 ± 7
Galangin	OH	H	OH	H	H	H	H	H	4 ± 9	-
Resokaempferol	H	H	OH	H	H	H	OH	H	38 ± 7	-
Kaempferol	OH	H	OH	H	H	H	OH	H	24 ± 8	-
Fisetin	H	H	OH	H	H	H	OH	OH	30 ± 13	-
Quercetin	OH	H	OH	H	H	OH	OH	H	48 ± 9	23 ± 5
Morin	OH	H	OH	H	OH	H	OH	H	38 ± 11	-
Herbacetin	OH	H	OH	OH	H	H	OH	H	77 ± 6	32 ± 8
Myricetin	OH	H	OH	H	H	OH	OH	OH	32 ± 12	-
Gossypetin	OH	H	OH	OH	H	OH	OH	H	65 ± 7	40 ± 8
Naringenin	-	-	-	-	-	-	-	-	−11 ± 5	-
Ketoconazole (positive control)	-	-	-	-	-	-	-	-	-	1.1 ± 0.4

**Table 2 biomedicines-13-00655-t002:** Information on the literature about the most active flavonoids.

Flavonoid	IC_50_ (μmol/L)	Substrate	Enzyme	Positive Control	Technique	Ref.
Baicalein	12.03	TST	RLM		HPLC	[28]
	9.2	BFC	BCS	KTZ	Fluorescence	[29]
	7.56–26.35	different substrates	HLM		Fluorescence	[5]
	9.60 ± 1.18	TST	HLM	KTZ	HPLC-UV	[30]
	15 ± 5	BOMR	BCS	KTZ	Fluorescence	This work
Luteolin	57.1 ± 16.1	BOMCC		ERYT	Fluorescence	[31]
	6.8	TST	HLM	KTZ	RP-HPLC	[4]
	4.62 ± 1.26	TST	HLM	KTZ	HPLC-UV	[30]
	31.2 ± 10.4	BOMR	BCS	KTZ	Fluorescence	This work
Herbacetin	<10		HLM		UPLC-MS/MS	[27]
	31.5 ± 8.0	BOMR	BCS	KTZ	Fluorescence	This work
Quercetin	16.7 ± 2.6; 3.03 ± 1.84	RPG	RLM; HLM	KTZ	HPLC	[32]
	28.0 ± 5.2	BOMCC	BCS	Erythro	Fluorescence	[31]
	4.3 ± 0.04	MDZ	HLM	KTZ	LC-MS/MS	[33]
	208.65	FLP	HLM	VPM	UPLC	[11]
	82 and 41	QNN	HLM		RP-HPLC	[34]
	4.1 ± 0.4	TZL	HLM	KTZ	HPLC	[35]
	22.1	TST	HLM	KTZ	RP-HPLC	[4]
	5.74 ± 1.16	TST	HLM	KTZ	HPLC-UV	[30]
	22.8 ± 5.2	BOMR	BCS	KTZ	Fluorescence	This work
Gossypetin	40.1 ± 7.7	BOMR	BCS	KTZ	Fluorescence	This work
Scutellarein	19.1 ± 7.1	BOMR	BCS	KTZ	Fluorescence	This work

**Table 3 biomedicines-13-00655-t003:** The binding energy predicted by the in silico docking of test flavones and controls to the CYP3A4 target. A, B, and C denote the ring system nearest the heme.

	Binding Energy (kcal/mol) and Ring System Closer to Heme (Ring)																										
	Pose Nr 1		Pose Nr 2			Pose Nr 3			Pose Nr 4			Pose Nr 5				Pose Nr 6			Pose Nr 7			Pose Nr 8			Pose Nr 9		
Baicalein	−8.40	B	−7.90		B	−7.80		AC	−7.80		AC	−7.80		AC		−7.60	AC		−7.60		B	−7.40		B	−7.30		AC
Herbacetin	−8.00	AC	−7.60		AC	−7.50		B	−7.50		AC	−7.40		B		−7.20	AC		−7.20		B	−7.00		B	−7.00		B
Luteolin	−8.80	AC	−8.70		AC	−8.60		B	−8.20		B	−7.90		AC		−7.90	AC		−7.70		B	−7.70		B	−7.30		AC
	**Binding Energy (kcal/mol) for Some Selected Antifungal and Antiviral Drugs**																										
	**AMD**	**AMP**		**CIMET**			**DAR**			**DILT**			**FLUCON**		**INDIN**			**KTZ**		**MICON**			**NEFAZ**			**VERAP**	
Controls	−8.6	−8.5		−6.2			−9.2			−7.7			−7.3		−10.9			−10.8		−8.2			−8.2			−7.4	

## Data Availability

The raw data supporting the conclusions of this article will be made available by the authors upon reasonable request.

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
