# Peer review of "How Plant Polyhydroxy Flavonoids Can Hinder the Metabolism of Cytochrome 3A4"

_biomedicines, 2025, doi:10.3390/biomedicines13030655_

Round 1

Reviewer 1 Report

Comments and Suggestions for Authors

The article deals with extensive studies on exploring the energetics of How polyhydroxy flavonoids hindering the metabolism 2 of Cytochrome 3A4
1.  The preferred domain of  the enzyme for which the guest molecules reside is not provided.

2.Further, detailed information on the binding sites, should be mentioned for each derivative of the flavonoid should be clearly explained.
3. Were there any unfavourable interactions existing between the host and guest complex?
4.Which forces were predominant in the binding stability  and what is the nature of the amino acid that was into bi nding, whether polar amino acids were more involved or non-polar,  
5.Were all the energy parameters, like intermolecular energy, electrostatic, etc., considered in
determining the docking score?

  1. Had the authors observe any unfavourable energy contributions
    7.What is the role of amino acid contribution? Does hydrophilic or hydrophobic amino acids govern the binding stability. Which is the most preferred domain?

8.How many conformers were generated for obtaining the stable docking score, the authors          have not discussed regarding the contour mapping of hydrogen-bonding and hydrophobic interactions existing between the flavonoids with the enzyme, since there seems to be several HB interactions through the OH moieties, also provide an extended table regarding the number of Conventional HB interactions along with non-conventional HB interactions, hydrophobic interactions and weaker forces of interaction.

  1.   
    Certain references regarding the docking of the enzyme with guest molecules that are published recently are to be incorporated.
  2. Fig:1, Colour representation to be provided for R groups substituted in the flavonoid structures, it would provide an easier understanding for the readers and also provide an explanation on the basis on the number of OH moieties influencing the energetics. How do you correlate based on the number of active hydrogen-bonding sites?
  3. Provide the inhibitor structures in the supporting information
  4. Please remove the flavonoid structures from Table 1.
  5. Table2: Provide the abbreviations separately as footnotes or at the end of the manuscript.
  6. References regarding globular proteins with flavonoids through docking studies have provided many break through regarding the binding amino acids and energetics pattern. The authors have missed out certain references pertaining to flavonoids in this context which are found to be essential regarding docking studies. If the authors are to revise the manuscript, they should consider the studies of flavanoids as reported in Journal of Chemical Health Risks JCHR (2024) 14(6), 858-878, (2024) 14(6), 858-878, Journal of Chemical Health Risks JCHR (2024) 14(5), 1505-1529, JCHR (2024) 14(6), 292-314

A major revision is required for publication; in the present form, it is not suitable

Reviewer 2 Report

Comments and Suggestions for Authors

This is a routine work about the interaction between protein and  chemicals.  I think the authors provide valuable data and should be published.  Some comments in detail:

  1. Fig 1 can be removed because the same information is included in Tab 1.
  2. Tab 2. Different substrates results in quite different results, the IC50 values are not of the same order of magnitude...  Can the authors provide some concrete explanation about this for general readers? By the way, the full name for BOMR is not introduced.
  3. Docking part.  Is the exhaustiveness big enough because the default value 8 may not a very good choice for a serious research.
  4. What force field (scoring function) did you use for the docking? It is a necessary information of the computation.  On contrast, you include too much technical details in the manuscript like the format of the middle-stage files.
  5. In the introduction part, I think you can write it more seriously.  E.g., I do not think "Consumers are seeking more “biological” foods and many edible natural products
    containing flavonoids" is a good statement for a scientific paper because almost all the foods (of course we have some exception like NaCl) are "biological".  I know the concept of "biological foods" but I think it has nothing to do with the topic here, and personally (I mean of course you can disagree with me) I do not like the way people use the word "biological" because it is misleading. If you really mean it, I think "eco-friendly" is a more clean way to say it, but again it is irrelevant. In this manuscript, we are discuss the problem at molecular level.

Round 2

Reviewer 1 Report

Comments and Suggestions for Authors

The authors have done extensive revision based on the queries , very satisfactory, the article can be accepted for publication